# EFFICIENT LONG SEQUENCE MODELING VIA STATE SPACE AUGMENTED TRANSFORMER

## ABSTRACT

Transformer models have achieved superior performance in various natural language processing tasks. However, the quadratic computational cost of the attention mechanism limits its practicality for long sequences. There are existing attention variants that improve the computational efficiency, but they have limited ability to effectively compute global information. In parallel to Transformer models, state space models (SSMs) are tailored for long sequences, but they are not flexible enough to capture complicated local information. We propose SPADE, short for **S**tate s**P**ace **A**ugmente**D** Transform**E**r. Specifically, we augment a SSM into the bottom layer of SPADE, and we employ efficient local attention methods for the other layers. The SSM augments global information, which complements the lack of long-range dependency issue in local attention methods. Experimental results on the Long Range Arena benchmark and language modeling tasks demonstrate the effectiveness of the proposed method. To further demonstrate the scalability of SPADE, we pre-train large encoder-decoder models and present fine-tuning results on natural language understanding and natural language generation tasks. Our code and pre-trained model checkpoints will be publicly available.

## 1 INTRODUCTION

With the rise of large language models, the difficulties of modeling long sequences have gained increasing attention. For instance, ChatGPT is capable of handling context that comprises up to 8k tokens, while GPT-4 (OpenAI, 2023) scales this ability up to 32k tokens. Conventional Transformer-based models rely on the attention mechanism (Vaswani et al., 2017), which computes a dependency score for every pair of tokens in the input sequence. Thus, full attention has a quadratic time and space complexity with respect to the length of the sequence. However, such complexity proves computationally prohibitive for tasks that require modeling long sequences such as text summarization (Nallapati et al., 2016) and question-answering (Kwiatkowski et al., 2019). In fact, we find that training a Transformer model (250M parameters) takes up over 80G of GPU memory when modelling an input sequence of length 8k.

In addition, Transformer models that rely on full attention run the risk of overfitting due to the lack of structural biases (Lin et al., 2022). The attention mechanism does not impose any structural prior over the input. As a result, order information (such as sinusoidal encoding) is required to train a Transformer model successfully. Transformer models, equipped with full attention mechanism, prove overly flexible resulting in overfitting to the noise. This significantly impacts the models' practicality in long sequence modeling, where the dependency signal is often weak, and signal-to-noise ratio is low (i.e., in long sequences, a majority of input tokens are useless). Empirical evidence shows that Transformer models without structural biases have a classification accuracy rate of 57.5% on a two-way classification task, nearly 30% less than state-of-the-art approaches that are equipped with powerful structural biases (see Section 4.1 for details).

Several methods have been proposed to address the problem of full attention's quadratic complexity and the need to introduce structural biases to Transformer models. One of the primary approaches is employing *approximation methods*, which approximates full attention using fast algorithms with linear complexity. Low-rank approximations (Wang et al., 2020b) or kernel methods (Peng et al., 2021), for instance, could simplify and speed up calculation of the attention score matrix (i.e.,

softmax($\mathbf{Q}\mathbf{K}^\top/\sqrt{d}$) in Eq. 1). Nevertheless, although these methods mitigate the computational complexity of full attention, they inherit the problems associated with the lack of structural bias.

To incorporate structural biases into Transformer, *partial attention* methods have been proposed. The methods can be further categorized in to *sparse attention* (Beltagy et al., 2020) and *clustering* (Kitaev et al., 2020). In the former approach, individual tokens attend to only a subset of tokens determined by pre-defined sparsity patterns. In the latter, tokens are divided into clusters and intra-cluster attention is performed. However, introducing these structural biases restricts the models' ability to capture global information. Local window attention, for instance, assumes that each token depends only on its direct neighbors, causing long-range and global information to be lost.

State space models (SSMs) introduce structural biases tailored for computing global information (Gu et al., 2022), in contrast to partial attention. Specifically, SSMs design fixed global dependency patterns that facilitate effective and efficient computation, and can be seen as linear recurrent neural networks with specifically designed fixed weights. Moreover, efficient algorithms have been developed to train these models. However, it should be noted that SSMs can be restrictive, because they do not capture local information as effectively as attention-based models, which explicitly compute dependencies among input tokens.

We propose SPADE, short for **S**tate s**P**ace **A**ugmente**D** Transform**E**r. SPADE is a multi-layer Transformer model that can effectively and efficiently capture complicated dependencies. Specifically, we augment a state space model (SSM) into the bottom layer of the model to integrate inputs with global information. Because the SSM only provides coarse global information, at the subsequent top layers of SPADE, we employ local attention methods to capture more complicated and refined local information. With this approach, the SSM induces a strong structural bias that augments global information and complements the long-range dependency issue in local attention methods.

We demonstrate the efficiency and effectiveness of SPADE on various tasks. First, we show that SPADE outperforms existing approaches on the Long Range Arena (Tay et al., 2021b) benchmark, which is designed to test models' ability in modeling long sequences. Second, we show that in autoregressive language modeling, SPADE is not only significantly faster than the vanilla Transformer (Vaswani et al., 2017), but also yields better performance. Third, we demonstrate the scalability of SPADE by conducting language model pre-training and fine-tuning experiments. Specifically, we pre-train an encoder-decoder model similar to T5 (Raffel et al., 2020). And we fine-tune the model on various tasks, including natural language understanding and natural language generation benchmarks. In all the settings, SPADE outperforms the baselines. Finally, we provide analysis and ablation experiments to further analyze and demonstrate the effectiveness of the proposed method.

## 2 BACKGROUND

### 2.1 ATTENTION MECHANISM

Suppose the input to the layer is $\mathbf{X} \in \mathbb{R}^{L \times d}$, where $L$ is the sequence length and $d$ is the embedding dimension, then the attention mechanism outputs

$$\text{Attn}(\mathbf{X}) = \text{softmax}\left(\frac{\mathbf{Q}\mathbf{K}^\top}{\sqrt{d}}\right)\mathbf{V}, \text{ where } \mathbf{Q} = \mathbf{X}\mathbf{W}_q, \ \mathbf{K} = \mathbf{X}\mathbf{W}_k, \ \mathbf{V} = \mathbf{X}\mathbf{W}_v. \qquad (1)$$

Here $\mathbf{W}_q, \mathbf{W}_k, \mathbf{W}_v \in \mathbb{R}^{d \times d}$ are learnable weights. The attention mechanism can simultaneously compute the alignment between any pair of input tokens, such that it models long-range dependencies better than recurrent neural networks. Specifically, denote the attention score matrix $\mathbf{A} = \text{softmax}(\mathbf{Q}\mathbf{K}^\top/\sqrt{d}) \in \mathbb{R}^{L \times L}$. Then, $\mathbf{A}_{ij}$ captures the alignment between the $i$-th and the $j$-th input tokens.

### 2.2 STATE SPACE MODELS

**Continuous time state space model.** A continuous time latent state space model maps a 1-dimensional input signal $u(t)$ to a $d_s$-dimensional latent state $x(t)$, after which $x(t)$ is mapped to a 1-dimensional output signal $y(t)$. Concretely,

$$x'(t) = \mathbf{A}x(t) + \mathbf{B}u(t), \quad y(t) = \mathbf{C}x(t). \qquad (2)$$

Here, $\mathbf{A} \in \mathbb{R}^{d_s \times d_s}$, $\mathbf{B} \in \mathbb{R}^{d_s}$ and $\mathbf{C} \in \mathbb{R}^{d_s}$.

Existing works leverage Eq. 2 to model long sequences. For example, Gu et al. (2020) claim that randomly initialized parameters $\mathbf{A}$, $\mathbf{B}$ and $\mathbf{C}$ cannot model long-range dependencies well. Subsequently, a class of matrices (termed HiPPO, high-order polynomial projection operators) are proposed to initialize $\mathbf{A}$. The HiPPO matrices are designed such that the state $x(t)$ at time $t$ can memorize the history of the input $u(t)$ up to time $t$.

**Discrete time state space model.** In practice, we often work with discrete sequences such as natural language inputs $(u_0, u_1, \cdots, u_L)$, where $L$ is the sequence length. To facilitate modeling discrete data, the model in Eq. 2 can be discretized (using the bilinear method) by a step size $\Delta$, such that

$$x_k = \overline{\mathbf{A}}x_{k-1} + \overline{\mathbf{B}}u_k, \quad y_k = \overline{\mathbf{C}}x_k, \tag{3}$$

where $\overline{\mathbf{A}} = (\mathbf{I} - \Delta/2 \cdot \mathbf{A})^{-1}(\mathbf{I} + \Delta/2 \cdot \mathbf{A})$, $\overline{\mathbf{B}} = (\mathbf{I} - \Delta/2 \cdot \mathbf{A})^{-1}\Delta\mathbf{B}$, $\quad \overline{\mathbf{C}} = \mathbf{C}$.

We unroll the above recurrent representation, after which we have

$$y_k = \overline{\mathbf{C}}\overline{\mathbf{A}}^k\overline{\mathbf{B}}u_0 + \cdots + \overline{\mathbf{C}}\overline{\mathbf{A}}\overline{\mathbf{B}}u_{k-1} + \overline{\mathbf{C}}\overline{\mathbf{B}}u_k.$$

This can be written as a convolutional representation

$$y = \overline{\mathbf{K}} * u, \text{ where } \overline{\mathbf{K}} \in \mathbb{R}^L = \left(\overline{\mathbf{C}}\overline{\mathbf{B}}, \overline{\mathbf{C}}\overline{\mathbf{A}}\overline{\mathbf{B}}, \cdots, \overline{\mathbf{C}}\overline{\mathbf{A}}^{L-1}\overline{\mathbf{B}}\right). \tag{4}$$

Here, $\overline{\mathbf{K}}$ is the convolutional kernel, "$*$" is the discrete convolution operator, $u$ represents the input sequence $(u_0, u_1, \cdots, u_L)$, and $y$ represents the corresponding output sequence $(y_0, y_1, \cdots, y_L)$.

In Eq. 4, the output $y$ can be computed efficiently given that the convolution kernel $\overline{\mathbf{K}}$ is known (e.g., using Fast Fourier Transform). However, computing the kernel is non-trivial. Most of existing algorithms have $O(L^2)$ time and space complexity.

**Structured State Space Sequence model (S4).** Gu et al. (2022) develop the S4 model to efficiently compute Eq. 4. Specifically, $\mathbf{C}$ in Eq. 2 is randomly initialized, and $\mathbf{A}$ and $\mathbf{B}$ are initialized as

$$\mathbf{A} = \mathbf{A}^{(d_s)} - \mathbf{P}\mathbf{P}^\top, \quad \mathbf{B}_i = (2i+1)^{1/2}, \tag{5}$$

where $\mathbf{P}_i = (i+1/2)^{1/2}$, $\mathbf{A}_{ij}^{(d_s)} = - \begin{cases} (i+1/2)^{1/2}(j+1/2)^{1/2}, & i > j, \\ 1/2, & i = j, \\ -(i+1/2)^{1/2}(j+1/2)^{1/2}, & i < j. \end{cases}$

Subsequently, the convolution kernel $\overline{\mathbf{K}}$ in Eq. 4 can be computed efficiently with near linear time and space complexity. Then, for an input $u$, the S4 output $y = \overline{\mathbf{K}} * u$ can be computed efficiently.

## 3 METHOD

We first conduct experiments to demonstrate that SSMs do not model local information well. Then, we present SPADE, which effectively combines global and local information by augmenting SSMs into the Transformer architecture.

### 3.1 ATTENTION VS. STATE SPACE MODELS

The motivation behind SPADE is that even though SSMs perform well on several long sequence classification tasks (Gu et al., 2022), they perform poorly on language modeling, which is a fundamental task in natural language processing. To demonstrate such an observation, we compare S4 with Transformer with full attention and Transformer with local (window and chunk) attention. In local attention, each token can only attend to its neighboring tokens (see Figure 1 for illustrations). We conduct experiments on token-level language modeling. In this setting, local information is more important than global information. This is because in practice, we rarely see words (tokens) that are thousands of positions apart exhibit strong dependencies (Sukhbaatar et al., 2019).

Experimental results are illustrated in Figure 2. We see that both Transformer with full attention and Transformer with local attention (e.g., window and chunk) outperforms S4. Notice that replacing

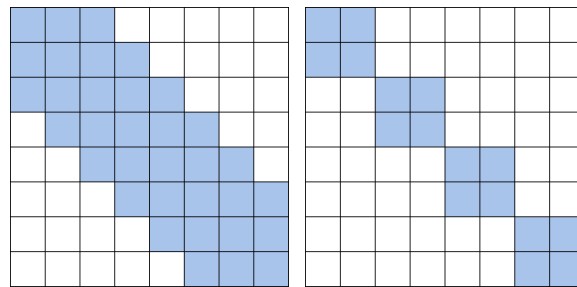

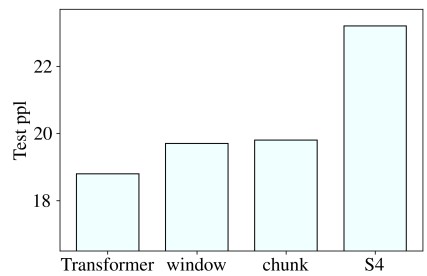

Figure 1: Illustration of window attention (left) and chunk attention (right). For window attention, the window size is 2 (on each side); for chunk attention, the chunk size is 2.

Figure 2: Performance of Transformer with full attention, window attention, chunk attention, and S4. We use language modeling experiments (see Section 4.2), and the sequence length is 3k.

full attention with local attention does not significantly hurt model performance, indicating that local information is more important in this setting. We remark that SSMs such as S4 produces a fixed dependency pattern, e.g., the convolution kernel in Eq. 4. Moreover, unlike attention, SSMs do not explicitly compute dependencies among tokens. Therefore, SSMs are not refined enough to capture local information, such that they perform poorly on language modeling tasks.

## 3.2 SPADE: STATE SPACE AUGMENTED TRANSFORMER

We propose SPADE, which is a multi-layer Transformer model that can capture complicated global and local information. The overall architecture of SPADE is shown in Figure 3 (left). The proposed model employs a hierarchical structure. Specifically, at the bottom layer of SPADE (termed the *global* layer), we capture global dependencies using a SSM. Because the SSM only provides coarse global information, the subsequent *local* layers facilitate the model to handle more refined and complicated local dependencies. In other words, the SSM induces a strong structural bias that augments global information to the inputs.

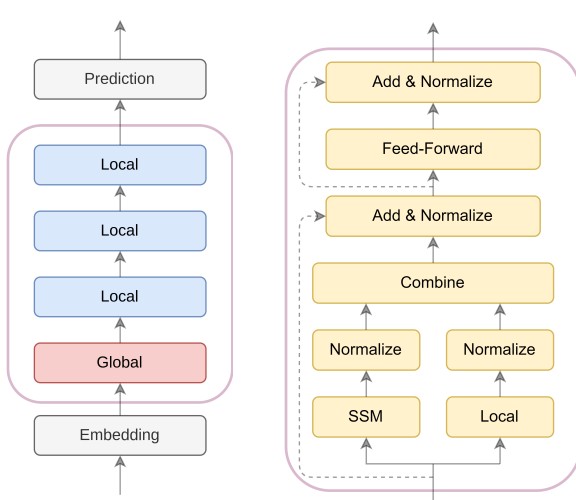

To instantiate the local layer, we replace the full attention in the conventional Transformer layer with off-the-shelf efficient local attention methods. SPADE is flexible to accommodate different approaches, such as window attention and chunk attention (see Figure 1 for illustrations).

Figure 3: Demonstration of SPADE with 4 layers. Left: model overview; Right: details of the global layer.

In the global layer (Figure 3, right), given the input $\mathbf{X}$ to the layer, we have the output $\mathbf{Y}$ as

$$\mathbf{Y} = \text{FFN}\left(\text{LN}(\mathbf{X}_a)\right) + \mathbf{X}_a, \text{ where } \mathbf{X}_a = \mathbf{W}\left[\text{LN}(\mathbf{X}_{\text{local}}), \text{LN}(\mathbf{X}_{\text{global}})\right] + \mathbf{X},$$
$$\mathbf{X}_{\text{local}} = \text{Local}\left(\text{LN}(\mathbf{X})\right), \ \mathbf{X}_{\text{global}} = \text{SSM}\left(\text{LN}(\mathbf{X})\right).$$

Here, $\text{LN}(\cdot)$ denotes layer normalization (Ba et al., 2016), $\text{FFN}(\cdot)$ denotes a two-layer feed-forward neural network, and $\mathbf{W}$ is a trainable weight that combines local and global representations. Notice that we apply normalization to $\mathbf{X}_{\text{local}}$ and $\mathbf{X}_{\text{global}}$ to align their scales. In this work, we choose S4 as the state space model.

We remark that because of the sequential nature of SSMs (Eq. 3), the global layer can encode positional information of the inputs. Therefore, we do not need additional fixed-length positional

embedding techniques (Devlin et al., 2019). Such a property enables SPADE to extrapolate to longer sequence length during testing, e.g., we can train a model with sequence length 512 and test the model with sequence length 1k.

## 4 EXPERIMENTS

In the experiments, we implement all the models using *PyTorch* (Paszke et al., 2019) and *Fairseq* (Ott et al., 2019). Training details such as hyper-parameter settings are deferred to the appendix.

### 4.1 LONG RANGE ARENA

**Dataset and models.** We evaluate SPADE on Long Range Arena (LRA, Tay et al. 2021b), which is a benchmark tailored for evaluating models' ability in modeling long sequences. Dataset details are presented in Appendix A.

Following the setting in Ma et al. 2022, we use small models (less than 2M parameters) for all the tasks. We limit the computational budget such that all the models are trained with similar speed for the same amount of time. To aggregate local information, we consider two approaches: window attention and chunk attention. For window attention, we sparsify the conventional softmax attention (termed *window*); and for chunk attention, we sparsify MEGA (Ma et al., 2022), which employs a gated attention technique (termed *chunk*).

**Results.** Experimental results are summarized in Table 1. We see that both variants of SPADE (window and hunk) significantly outperform all the baselines in terms of average accuracy. For example, the window attention variant outperforms the best-performing baseline (MEGA-chunk) by 0.5%, and the chunk attention variant has a 1.8% performance gain. Therefore, SPADE is more suitable to model long sequences than existing approaches.

Table 1: Experimental results on LRA. For Path-X, "✗" indicates unavailable results due to computational constraints. See Table 11 in the appendix for comparison with other baselines.

| Dataset | Listops | Text | Retrieval | Image | Pathfinder | Path-X | Avg. |
|---|---|---|---|---|---|---|---|
| Sequence length | 2k | 4k | 8k | 1k | 1k | 16k | — |
| Transformer (full) | 36.37 | 64.27 | 57.46 | 42.44 | 71.40 | ✗ | 53.66 |
| S4 (Gu et al., 2022) | 58.35 | 76.02 | 87.09 | 87.26 | 86.05 | 88.10 | 80.48 |
| MEGA-chunk (Ma et al., 2022) | 58.76 | 90.19 | 90.97 | 85.80 | 94.41 | 93.81 | 85.66 |
| SPADE (window) | 59.70 | 87.55 | 90.13 | **89.11** | **96.42** | 94.22 | 86.19 |
| SPADE (chunk) | **60.50** | **90.69** | **91.17** | 88.22 | 96.23 | **97.60** | **87.40** |

### 4.2 LANGUAGE MODELING

We further evaluate our model by conducting language modeling experiments on Wikitext-103. The dataset contains English-language Wikipedia articles, and the total number of tokens is 103M. In all the experiments, we follow the settings in Baevski & Auli (2019), where we use a large-scale Transformer model with 16 layers and about 250M parameters. We set the input sequence length to 3k and train for 286k steps. During testing, we set the sequence length to 3k and the context window size to 400. Similar to the LRA experiments, we equip SPADE with either window attention (*window*) or chunk attention (*chunk*).

Table 2: Experimental results on Wikitext-103.

| | Test ppl |
|---|---|
| Transformer (Vaswani et al., 2017) | 18.8 |
| Transformer (window) | 19.7 |
| S4 (Gu et al., 2022) | 23.2 |
| FLASH-chunk (Hua et al., 2022) | 20.9 |
| MEGA-chunk (Ma et al., 2022) | 19.8 |
| SPADE (chunk) | 19.5 |
| SPADE (window) | **18.5** |

Experimental results are presented in Table 2. From the results, we see that by combining global and local information, the proposed model achieves significant performance improvement and out-

perform all the baselines. For example, the vanilla window attention has a 19.7 perplexity on the test set, and by integrating a SSM into SPADE, we achieve a 1.2 perplexity gain. We remark that SPADE with window attention is not only significantly faster than the Transformer with full attention, but also yields a better performance.

**Remark.** We remark that we do not need to train the S4 in the bottom layer of SPADE to achieve the performance in Table 2. That is, we initialize the parameters in S4 using Eq. 5, and the parameters are frozen during training. This is because even without training, the initialization of S4 yields intriguing theoretical properties, which facilitates S4's ability to capture global information.

## 5 LANGUAGE MODEL PRE-TRAINING

We implement model pre-training using *Fairseq*, and we implement model fine-tuning using *MT-DNN* (Liu et al., 2019a; 2020b). Note that all our experiments only use single task fine-tuning. Details such as pre-training settings and hyper-parameter settings are deferred to the appendix.

### 5.1 PRE-TRAINING DETAILS

To demonstrate the scalability of the proposed method, we pre-train an encoder-decoder variant of SPADE. The model architecture is the same as T5$_{base}$ (Raffel et al., 2020), except that we use post-layernorm instead of pre-layernorm to improve model performance (Liu et al., 2020a; Xiong et al., 2020). The embedding dimension is 768, the hidden dimension of the FFN is 3072, the number of attention heads is 12, and both the encoder and the decoder have 12 layers. We add a S4 module to the bottom layer of SPADE, and the parameters of the S4 are fixed after initialization (Eq. 5). We use the window attention as the local information extractor, where we set the window size to 128. The model contains about 290M parameters.

We consider two pre-training settings. In the first setting, we follow the pre-training settings in BERT (Devlin et al., 2019), and we term the trained model **SPADE$_{base}$**. In the second setting, we follow the pre-training settings in RoBERTa (Liu et al., 2019b), and we term the trained model **SPADE$_{base++}$**.

We remark that because the S4 module is not trained after proper initialization, and we do not use fixed-length positional embedding, our pre-trained model can extrapolate to any sequence length. For example, we can set the sequence length to 16k during fine-tuning, which is longer than the sequence length used in pre-training.

### 5.2 NATURAL LANGUAGE UNDERSTANDING

We fine-tune the pre-trained models on the General Language Understanding Evaluation (GLUE) benchmark (Wang et al., 2019), which is a collection of natural language understanding tasks. Dataset details are presented in Appendix A. We do not consider the long sequence setting in these tasks. In the experiments, all models are fine-tuned under the sequence length of 512.

Table 3: Experimental results on GLUE development set. *T5$_{base}$* results are from Raffel et al. 2020.

| | RTE Acc | MRPC Acc/F1 | CoLA Mcc | SST-2 Acc | STS-B P/S Corr | QNLI Acc | QQP Acc/F1 | MNLI-m/mm Acc | Avg. Score |
|---|---|---|---|---|---|---|---|---|---|
| T5$_{base}$ | 76.9 | 90.8/– | 55.5 | 92.8 | 86.5 | 91.9 | 90.9/– | 84.4/83.5 | — |
| T5$_{base}$ (re-imp) | 78.0 | 91.7/88.6 | 61.5 | 93.6 | 88.2 | 92.9 | 91.2/87.9 | 87.0/86.9 | 85.1 |
| SPADE$_{base}$ | 77.9 | 92.2/89.0 | 63.2 | 94.0 | 87.9 | 92.8 | 91.6/88.2 | 87.1/87.2 | 85.4 |
| SPADE$_{base++}$ | **80.5** | **92.3/89.2** | **64.7** | **95.9** | **89.2** | **93.9** | **91.7/88.4** | **89.6/89.2** | **86.8** |

Experimental results are presented in Table 3. We see that both variants of SPADE significantly outperforms T5$_{base}$. For example, T5$_{base}$ has a 83.9 average accuracy on the MNLI dataset (average of MNLI-m and MNLI-mm); while SPADE$_{base}$ has a 87.2 average accuracy (+3.3) and SPADE$_{base++}$ has a 89.4 average accuracy (+5.5). Recall that the sequence length is set to 512, which is the standard setting instead of the long-sequence setting. Therefore, the results indicate that SPADE is universal in that it is suitable to model both long and short sequences.

## 5.3 Natural Language Generation

We also fine-tune the pre-trained models on several abstractive summarization datasets. Dataset details are presented in Appendix A. We use ROUGE-2 scores as the evaluation metric.

We compare SPADE with LongT5 (Guo et al., 2022), which is a state-of-the-art model tailored for long sequences. Experimental results are summarized in Table 4. From the results, we see that our model significantly outperforms LongT5. Note that SPADE$_{base++}$ have about 290M parameters, while LongT5$_{large}$ contains about 770M parameters and LongT5$_{xl}$ contains about 3B parameters. From the results, we see that in all the tasks, our base-sized models have on par or better performance compared with LongT5$_{large}$. On the MultiNews dataset, our model even outperforms LongT5$_{xl}$, which is over ten times larger than our model.

Table 4: Experimental results (ROUGE-2) on test sets. LongT5 results are from Guo et al. 2022.

| | arXiv | | CNN/DailyMail | |
| --- | --- | --- | --- | --- |
| | Length | R-2 | Length | R-2 |
| LongT5$_{base}$ | 4k | 18.54 | 4k | 20.11 |
| LongT5$_{large}$ | 16k | 21.63 | 4k | 20.51 |
| LongT5$_{xl}$ | 16k | 21.92 | 4k | 21.40 |
| SPADE$_{base++}$ | 16k | 21.65 | 4k | 20.40 |
| | MediaSum | | MultiNews | |
| | Length | R-2 | Length | R-2 |
| LongT5$_{base}$ | 4k | 18.35 | 4k | 17.37 |
| LongT5$_{large}$ | 4k | 19.04 | 8k | 18.44 |
| LongT5$_{xl}$ | 4k | 19.66 | 8k | 19.43 |
| SPADE$_{base++}$ | 4k | 19.03 | 8k | 19.63 |

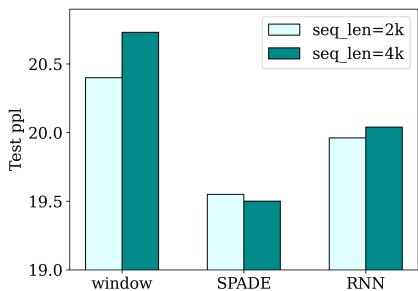

Figure 4: Performance of models with different global information extractors under two sequence length. We conduct language modeling experiments on Wikitext-103 with window attention (window=128).

## 6 Analysis

### 6.1 Effectiveness of Global Information Extractors

Recall from Eq. 3 that SSMs are essentially linear recurrent neural networks which can be computed much more efficiently than conventional RNNs. In Figure 4, we explore model variants with different global information extractors. Specifically, we equip Transformer with window attention with either S4 or LSTM (Hochreiter & Schmidhuber, 1997).

From the results, we see that RNNs can indeed capture global information. Also, we see that SSMs are more effective than RNNs. This is because RNNs suffer from known numerical problems such as forgetting and gradient explosion/vanishing (Pascanu et al., 2013). On the other hand, the theoretical properties of SSMs (Gu et al., 2020; 2021; 2022) can greatly alleviate such issues.

### 6.2 Efficiency Comparison

We compare the efficiency of SPADE with other models: Transformer with full attention, Transformer with window attention, MEGA-chunk, and S4. The results are illustrated in Figure 5. We see that SPADE is efficient in terms of both training speed and GPU memory usage. For example, when the sequence length is 6k, Transformer uses about 60GB of GPU memory, whereas SPADE with window attention only uses 27GB. Moreover, notice that SPADE also trains significantly faster than the vanilla Transformer under all settings. Notice that S4 may be less efficient than the vanilla Transformer (e.g., when the sequence length is 3k). This is because in Gu et al. 2022, each layer of the model contains multiple S4 modules and expensive non-linear components. Therefore, the per-layer computational cost can exceed full attention when the sequence is not extremely long.

We remark that even though we add a S4 module to the bottom layer of SPADE, such an additional module does not induce much computational overhead. We see that both the training speed and the

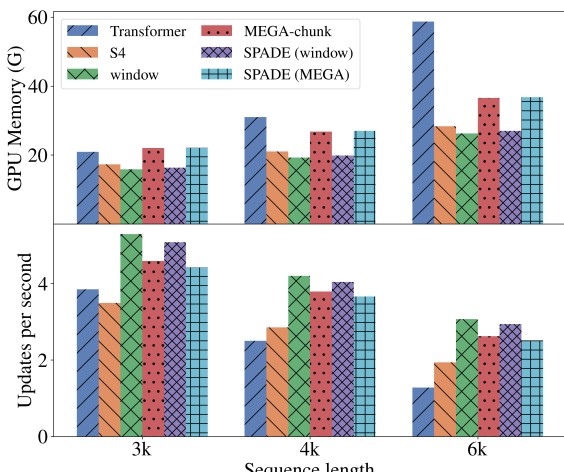

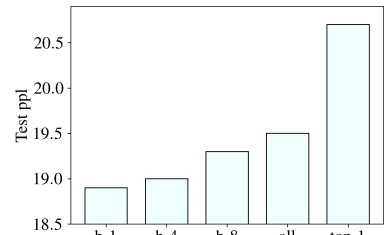

Figure 6: Performance vs. location of SSMs. We conduct language modeling experiments with window attention (window=256). By default, the model has 16 layers, where the bottom layer is a global layer and the rest are local layers. Here, "*b-k*" means the bottom-k layers are global layers, "*all*" means all layers are global layers, and "*top-1*" means the top-1 layer is a global layer.

Figure 5: Efficiency comparison of differnet models on language modeling tasks.

memory usage of SPADE with window is only marginally different from those of window-attention Transformer. We have similar observations for the chunk attention variant.

## 6.3 LOCATION AND NUMBER OF GLOBAL LAYERS

Recall that in SPADE, the bottom layer is equipped with a SSM and serves as the global layer, while the rest are local layers (see Figure 3). In Figure 6, we empirically justify this design choice.

We first investigate the possibility of incorporating more global layers: we set the bottom 1 (the default choice), 4, 8, and 16 (all) layers as global layers. From the results, we see that model performance decreases as we use more global layers. This is because the SSM in the bottom layer captures and filters out global information, such that subsequent SSMs only introduce noise to the intermediate representations.

We also investigate whether the global layer can be the top instead of the bottom layer in SPADE. From Figure 6, we see that model performance drops significantly. This is because as a global information extractor, the global layer encodes positional information, on which the local attention modules rely. Therefore, using the global layer as the top layer is akin to using Transformer models without positional encoding, which will yield unsatisfactory performance.

## 6.4 DIFFERENT CONFIGURATIONS

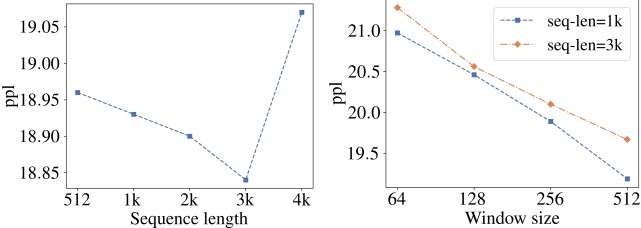

| | Sequence length | | | |
|---|---|---|---|---|
| | 2k | 3k | 4k | 6k |
| 128 | 19.55 | 19.42 | 19.50 | 19.55 |
| 256 | 18.90 | 18.95 | 18.99 | 19.00 |
| 512 | 18.63 | 18.64 | 18.52 | 18.67 |

Table 5: Performance of SPADE with window attention using different sequence length and window size. The first column is window size. We conduct language modeling experiments on Wikitext-103.

Figure 7: Performance with different configurations. Left: Transformer with full attention using different sequence length; Right: Transformer with window attention using different window size and sequence length.

We examine how performance changes when we change the sequence length and window size.

From Figure 7 (left), we see that when we increase the sequence length from 512 to 3k, performance of Transformer with full attention increases. However, when we further increase the sequence length

to 4k, model performance drastically drops. This is because in long sequences, the signal-to-noise ratio is low, such that the full attention may easily fit to the noise. From Figure 7 (right), we see that performance of Transformer with window attention increases when we increase the window size. Moreover, model performance is better with shorter sequences for the same window size. Such findings indicate that performance of window attention depends on the proportion of information within its perception.

From Table 5, we see that for the same sequence length, performance of SPADE increases when we increase the window size. Also, we see that performance of SPADE marginally decreases when we increase the sequence length from 4k to 6k. Recall from Figure 7 (left) that performance of Transformer with full attention drastically deteriorates when we increase the length from 3k to 4k. Such a result indicates that the proposed model is more suitable to model long sequences.

## 7 RELATED WORKS

In Eq. 1, we have $\mathbf{Q}, \mathbf{K}, \mathbf{V} \in \mathbb{R}^{L \times d}$, such that computing the attention $\text{Attn}(\mathbf{X})$ introduces $O(L^2)$ time and space costs. Such quadratic costs are prohibitive when the sequence length $L$ is large. There are various attempts to reduce the quadratic time and space complexity of the vanilla attention.

One approach is to employ *sparse attention*. That is, each token only attends to a subset of all the tokens according to pre-defined patterns, e.g., neighboring tokens within a fixed size window. Some examples include Sparse Transformer (Child et al., 2019), BlockBERT (Qiu et al., 2020), Longformer (Beltagy et al., 2020), ETC (Ainslie et al., 2020), BigBird (Zaheer et al., 2020), HEPOS (Huang et al., 2021), and Poolingformer (Zhang et al., 2021).

Another approach is to use *low-rank projection*. For example, in Linformer (Wang et al., 2020b), the attention mechanism in Eq. 1 becomes $\text{Attn}(\mathbf{X}) = \text{softmax}(\mathbf{Q}(\mathbf{EK})^\top / \sqrt{d})(\mathbf{FV})$. Here, the two additional parameters satisfy $\mathbf{E}, \mathbf{F} \in \mathbb{R}^{r \times L}$, where $r$ is the projection rank such that $r \ll L$. Similar methods include Nyströmformer (Xiong et al., 2021), Synthesizer (Tay et al., 2021a), Transformer-LS (Zhu et al., 2021a), and Luna (Ma et al., 2021). However, these approaches face difficulty when handling causal tasks, such as auto-regressive language modeling. Specifically, in Eq. 1, we mask out the upper triangular part in the attention score matrix $\mathbf{A} \in \mathbb{R}^{L \times L}$ such that each token can only attend to its previous tokens. However, this is implausible in Linformer since we project the $L \times L$ matrix to a $L \times r$ matrix.

*Kernel-based approaches* can be used to approximate the full attention $\text{Attn}(\mathbf{X})$. In these approaches, the quadratic-time softmax attention is replaced by fast linear-time kernel approximations (e.g., Gaussian and arc-cosine kernel). Some examples include Linear Transformer (Katharopoulos et al., 2020), Performer (Choromanski et al., 2021), Random Feature Attention (Peng et al., 2021), and FMMformer (Nguyen et al., 2021). Both low-rank projection and kernel-based approaches approximate the full attention, and thus, they often suffer from non-negligible approximation error.

We can also adopt *clustering-based approaches*, where we divide $\mathbf{Q}$ or $\mathbf{K}$ into several clusters, and only perform inter-cluster attention. Such methods include Reformer (Kitaev et al., 2020), Clusterformer (Wang et al., 2020a), Sinkhorn Transformer (Tay et al., 2020), Fast Transformer (Vyas et al., 2020), Routing Transformer (Roy et al., 2021), and FLASH (Hua et al., 2022).

## 8 CONCLUSION AND DISCUSSION

In this work, we propose SPADE, a state space augmented Transformer model that targets long sequence modeling. SPADE is a multi-layer Transformer model, where the bottom layer is a global layer and the rest are local layers. In the global layer, we use a SSM to augment coarse global information, which are subsequently refined by the following local layers. We instantiate the local layers with off-the-shelf efficient attention methods, such as window attention. The proposed model has linear time and space computationally complexity, facilitating it to handle long sequences. We conduct extensive experiments on the Long Range Arena (LRA) benchmark and language modeling datasets to demonstrate the effectiveness and efficiency of SPADE. We also pre-train encoder-decoder models to demonstrate the scalability of SPADE, and we perform fine-tuning experiments on natural language understanding (GLUE) and natural language generation (summarization) tasks. In all the experiments, SPADE exhibits superior performance and outperforms the baselines.

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

## A DATASET DETAILS

**Long Range Arena.** We evaluate the effectiveness of the proposed model on Long Range Arena (LRA, Tay et al. 2021b), which is a benchmark tailored for evaluating models' ability in modeling long sequences. The benchmark contains six tasks: ListOps, which tests the capability of modeling hierarchically structured data (Nangia & Bowman, 2018); byte-level text classification on the IMDB movie review dataset (Text, Maas et al. 2011); byte-level document retrieval on the ACL anthology network (Retrieval, Radev et al. 2013); pixel-level image classification on CIFAR-10 (Image, Krizhevsky et al. 2009); Pathfinder, which tests the capability in modeling spatial dependency (Linsley et al., 2018); and a longer version of Pathfinder (Path-X, Tay et al. 2021b).

**GLUE.** For natural language understanding, we fine-tune the pre-trained models on the General Language Understanding Evaluation (GLUE) benchmark (Wang et al., 2019), which is a collection of natural language understanding tasks. The benchmark includes two single-sentence classification tasks: CoLA (Warstadt et al., 2019) is a linguistic acceptability task; and SST-2 (Socher et al., 2013) is a binary classification task that classifies movie reviews to positive or negative. The benchmark also contains three similarity and paraphrase tasks: STS-B (Cer et al., 2017) is a text similarity task; MRPC (Dolan & Brockett, 2005) is a paraphrase detection task; and QQP is a duplication detection task. Additionally, there are natural language inference tasks: MNLI (Williams et al., 2018); QNLI (Rajpurkar et al., 2016); RTE (Dagan et al., 2006; Bar-Haim et al., 2006; Giampiccolo et al., 2007; Bentivogli et al., 2009). Statistics of the GLUE benchmark is summarized in Table 7.

**Summarization.** For natural language generation, we fine-tune the pre-trained models on several abstractive summarization datasets. The sources and statistics are summarized in Table 6.

Table 6: Statistics and sources of abstractive summarization datasets.

|  | # Train | # Validation | # Test | Mean | Median | Max | 90th percentile |
|---|---|---|---|---|---|---|---|
| arXiv (Cohan et al., 2018) | 203,037 | 6,436 | 6,440 | 10,720 | 8,519 | 378,825 | 20,170 |
| CNN/DailyMail (Nallapati et al., 2016) | 287,113 | 13,368 | 11,490 | 982 | 894 | 5,268 | 1,659 |
| MediaSum (Zhu et al., 2021b) | 443,596 | 10,000 | 10,000 | 2,302 | 1,748 | 125,974 | 4,128 |
| MultiNews (Fabbri et al., 2019) | 44,972 | 5,622 | 5,622 | 2,594 | 1,902.5 | 683,544 | 4,853 |

Table 7: Statistics of the GLUE benchmark.

| **Corpus** | Task | # Train | # Dev | # Test | # Labels | Metrics |
|---|---|---|---|---|---|---|
| Single-Sentence Classification | | | | | | |
| CoLA | Acceptability | 8.5k | 1k | 1k | 2 | Matthews corr |
| SST | Sentiment | 67k | 872 | 1.8k | 2 | Accuracy |
| Pairwise Text Classification | | | | | | |
| MNLI | NLI | 393k | 20k | 20k | 3 | Accuracy |
| RTE | NLI | 2.5k | 276 | 3k | 2 | Accuracy |
| QQP | Paraphrase | 364k | 40k | 391k | 2 | Accuracy/F1 |
| MRPC | Paraphrase | 3.7k | 408 | 1.7k | 2 | Accuracy/F1 |
| QNLI | QA/NLI | 108k | 5.7k | 5.7k | 2 | Accuracy |
| Text Similarity | | | | | | |
| STS-B | Similarity | 7k | 1.5k | 1.4k | 1 | Pearson/Spearman corr |

## B TRAINING DETAILS

### B.1 LANGUAGE MODEL PRE-TRAINING AND FINE-TUNING

For language model pre-training, we consider two pre-training settings:

◇ SPADE$_{base}$: We follow the pre-training settings in BERT (Devlin et al., 2019). Specifically, we train the model on Wikipedia (Devlin et al., 2019) and BookCorpus (Zhu et al., 2015).

◇ SPADE$_{base++}$: We follow the pre-training settings in RoBERTa (Liu et al., 2019b). Specifically, we train the model on Wikipedia (Devlin et al., 2019), BookCorpus (Zhu et al., 2015), STORIES (Trinh & Le, 2018), CC-News (Liu et al., 2019b), and OpenWebText (Gokaslan et al., 2019).

For language model pre-training and fine-tuning experiments, we use Adam (Kingma & Ba, 2015) as the optimizer. Hyper-parameters for pre-training are detailed in Table 8; and hyper-parameters for fine-tuning are detailed in Table 9.

Table 8: Hyper-parameters for pre-training.

| Parameters | Base | Base++ |
|---|---|---|
| Peak Learning Rate | 4e-4 | 2e-4 |
| Batch Size | 2,048 | 2,048 |
| Warmup Steps | 10,000 | 10,000 |
| Total Steps | 125,000 | 2,000,000 |
| Sequence Length | 1024 | 1024 |
| Relative Position Encoding Buckets | 32 | 32 |
| Relative Position Encoding Max Distance | 128 | 128 |
| Adam $\epsilon$ | 1e-6 | 1e-6 |
| Adam $(\beta_1, \beta_2)$ | (0.9, 0.98) | (0.9, 0.98) |
| Clip Norm | – | 1.0 |
| Dropout | 0.1 | 0.1 |
| Weight Decay | 0.01 | 0.01 |

Table 9: Hyper-parameters for fine-tuning.

| Parameters | Range |
|---|---|
| Learning Rate | {2e-5, 4e-5, 5e-5, 1e-4} |
| Batch Size | {16, 32} |
| Maximum Training Epochs | {3, 5, 10} |
| Dropout | 0.1 |
| Warmup Step Rate | 0.1 |
| Weight Decay | 0.1 |

## B.2 LONG RANGE ARENA

We follow the model architecture settings in Ma et al. 2022. In all the experiments, we use Adam (Kingma & Ba, 2015) as the optimizer. We use a linear decay learning rate schedule. The rest of the hyper-parameters are detailed in Table 10.

Table 11 shows additional experimental results, where we compare SPADE with existing efficient Transformer models.

## B.3 LANGUAGE MODELING

We follow the settings in Baevski & Auli 2019, including model architecture and hyper-parameters. For the efficient Transformer variants, we set the window size to 512 when using window attention, and we set the chunk size to 512 when using chunk attention.

Table 10: Hyper-parameters for training on LRA.

| Task | Batch Size | Learning Rate | Weight Decay | Dropout | Clip Norm | Chunk Size |
|------|-----------|---------------|--------------|---------|-----------|------------|
| Listops | 64 | 0.0015 | 0.0 | 0.2 | 1.0 | 128 |
| Text | 100 | 0.01 | 0.01 | 0.2 | 1.0 | 128 |
| Retrieval | 128 | 0.004 | 0.03 | 0.1 | 1.0 | 128 |
| Image | 100 | 0.01 | 0.02 | 0.0 | 1.0 | 128 |
| Pathfinder | 128 | 0.01 | 0.01 | 0.0 | 1.0 | 128 |
| Path-X | 16 | 0.01 | 0.01 | 0.0 | 1.0 | 1024 |

Table 11: Experimental results on Long Range Arena (LRA). Path-X uses 16k as the input sequence length, and "✗" indicates unavailable results due to computational constraints. All the baseline results, except for MEGA-chunk, are from Gu et al. (2022). MEGA-chunk results are from Ma et al. (2022).

| Dataset | Listops | Text | Retrieval | Image | Pathfinder | Path-X | Avg. |
|---------|---------|------|-----------|-------|------------|--------|------|
| Sequence length | 2k | 4k | 8k | 1k | 1k | 16k | — |
| Random | 10.00 | 50.00 | 50.00 | 10.00 | 50.00 | 50.00 | 36.67 |
| Transformer (full) (Vaswani et al., 2017) | 36.37 | 64.27 | 57.46 | 42.44 | 71.40 | ✗ | 53.66 |
| Transformer (window) | 15.82 | 52.98 | 53.39 | 41.46 | 66.63 | ✗ | 46.71 |
| Sparse Trans. (Child et al., 2019) | 17.07 | 63.58 | 59.59 | 44.24 | 71.71 | ✗ | 51.03 |
| Longformer (Beltagy et al., 2020) | 35.63 | 62.85 | 56.89 | 42.22 | 69.71 | ✗ | 52.88 |
| Linformer (Wang et al., 2020b) | 35.70 | 53.94 | 52.27 | 38.56 | 76.34 | ✗ | 51.14 |
| Reformer (Kitaev et al., 2020) | 37.27 | 56.10 | 53.40 | 38.07 | 68.50 | ✗ | 50.56 |
| Sinkhorn Trans. (Tay et al., 2020) | 33.67 | 61.20 | 53.83 | 41.23 | 67.45 | ✗ | 51.23 |
| Synthesizer (Tay et al., 2021a) | 36.99 | 61.68 | 54.67 | 41.61 | 69.45 | ✗ | 52.40 |
| BigBird (Zaheer et al., 2020) | 36.05 | 64.02 | 59.29 | 40.83 | 74.87 | ✗ | 54.17 |
| Linear Trans. (Katharopoulos et al., 2020) | 16.13 | 65.90 | 53.09 | 42.34 | 75.30 | ✗ | 50.46 |
| Performer (Choromanski et al., 2021) | 18.01 | 65.40 | 53.82 | 42.77 | 77.05 | ✗ | 51.18 |
| FNet (Lee-Thorp et al., 2022) | 35.33 | 65.11 | 59.61 | 38.67 | 77.80 | ✗ | 54.42 |
| Nystromformer (Xiong et al., 2021) | 37.15 | 65.52 | 79.56 | 41.58 | 70.94 | ✗ | 57.46 |
| Luna-256 (Ma et al., 2021) | 37.25 | 64.57 | 79.29 | 47.38 | 77.72 | ✗ | 59.37 |
| FMMformer (Nguyen et al., 2021) | 36.74 | 67.84 | 81.88 | 45.10 | 72.12 | ✗ | 60.74 |
| S4 (Gu et al., 2022) | 58.35 | 76.02 | 87.09 | 87.26 | 86.05 | 88.10 | 80.48 |
| MEGA-chunk (Ma et al., 2022) | 58.76 | 90.19 | 90.97 | 85.80 | 94.41 | 93.81 | 85.66 |
| SPADE (window) | 59.70 | 87.55 | 90.13 | **89.11** | **96.42** | 94.22 | 86.19 |
| SPADE (chunk) | **60.50** | **90.69** | **91.17** | 88.22 | 96.23 | **97.60** | **87.40** |

