# OpenReview forum: "Efficient Long Sequence Modeling via State Space Augmented Transformer"
_ICLR.cc/2024/Conference — Submitted to ICLR 2024_

### Official Review · Reviewer_w8HW · 2023-10-14

**Soundness:** 2 fair
**Presentation:** 3 good
**Contribution:** 2 fair
**Rating:** 3
**Confidence:** 4

**Summary:**

Authors propose to augment the Transformer with SSM models to achieve better performance from computational and memory consumption perspectives.

Experiments show that such architecture outperforms selected baselines on language modeling tasks and on Long Range Arena datasets. Furthermore, the authors showed that such a model could be successfully utilized for pre-training and fine-tuning, as shown in Section 5.

**Strengths:**

- The paper is well-motivated and solved task is important for the field.

- The paper is mostly well-written except for some flaws described in the weaknesses Section.

**Weaknesses:**

- My main concern for this paper is the lack of baselines. Including a comparison with other recent SSMs, such as S5 [1] or Hyena [2], would be highly beneficial.
- The abovementioned models could be incorporated with SPADE by replacing the SSM module with any other model. However, it would be beneficial to understand whether the performance gap between SPADE and S4 is caused by adding Transformer blocks on top of the SSM module. Training SPADE with Hyena may not improve performance over Hyena while being better than SPADE with S4. In this case, the motivation for the paper will disappear.
- I struggled to understand which SSM was used in SPADE (did I miss it?). It should be S4 based on the text. However, it would be helpful to name it in Section 3.2 explicitly.
- The paper lacks reproducibility despite an extensive description of training within the text since no supplementary material with source code was released.

[1] https://arxiv.org/pdf/2208.04933.pdf

[2] https://arxiv.org/pdf/2302.10866.pdf

While at this point, I voted for rejecting this paper, I believe that flaws with baselines could be fixed. Then, I would happily increase my score.

**Questions:**

Please refer to the weaknesses section

---

> ### Author Response · Authors · 2023-11-23
>
> Weaknesses:
>
> * We conduct additional experiments to demonstrate the effectiveness of SPADE. Specifically, we use S4, S5 [2], and Hyena [1] as the SSM in SPADE (Figure 3). We remark that other SSMs, such as GSS [3], are also outperformed by state-of-the-art models: S5 and Hyena. We also compare SPADE with positional embedding methods such as relative positional embedding and rotary positional embedding. We conduct language modeling experiments, and we report the ppl on the test set (lower the better). All models in the experiments have approximately 120M parameters.
>
> * From the results, we can see that SPADE is easily extendable to accommodate various SSM. Moreover, for all configurations, we see that SPADE significantly outperforms the SSM-only model.
>
> * In the experiments, we use the standard S4 as the SSM.
>
> * We have included the code in the supplemental material.
>
> |  | ppl |
> |---|---|
> | Transformer | 22.3 |
> | Transformer-Relative | 20.9 |
> | Transformer-Rotary | 20.8 |
> | S4 | 22.7 |
> | SPADE-S4 | 20.2 |
> | S5 | 23.0 |
> | SPADE-S5 | 20.0 |
> | Hyena | 22.9 |
> | SPADE-Hyena | 20.3 |
>
> [1] https://arxiv.org/pdf/2302.10866.pdf \
> [2] https://arxiv.org/pdf/2208.04933.pdf \
> [3] https://openreview.net/forum?id=5MkYIYCbva

---

### Official Review · Reviewer_iRvY · 2023-11-01

**Soundness:** 3 good
**Presentation:** 3 good
**Contribution:** 3 good
**Rating:** 5
**Confidence:** 3

**Summary:**

Summary:

The paper proposes SPADE (State sPace AugmenteD TransformEr), a novel approach for efficient long sequence modeling. SPADE integrates a state space model (SSM) into the bottom layer of a Transformer architecture, which enhances global information processing. This is complemented by local attention methods in the upper layers to handle local dependencies. The proposed method addresses the limitations of existing attention variants that struggle with long-range dependencies and computational inefficiency. SPADE demonstrates improved performance on the Long Range Arena benchmark and various language modeling tasks. The architecture allows SPADE to scale efficiently and outperform baselines in both natural language understanding and generation tasks, with the additional benefit of being able to handle longer sequences than it was trained on due to the SSM's extrapolation capabilities.

**Strengths:**

Advantages:

 - Integration of State Space Models: SPADE incorporates a state space model into the bottom layer of the architecture, providing a strong structural bias for augmenting global information and addressing long-range dependency issues present in local attention methods​.

 - Performance on Benchmarks: SPADE outperforms existing approaches (arguably marginally) on the Long Range Arena benchmark, specifically designed to assess models' ability to handle long sequences​.

 - Efficiency and Speed: In autoregressive language modeling tasks, SPADE is significantly faster and more performant than the vanilla Transformer model​.

**Weaknesses:**

Disadvantages:

 - Incremental Performance Gain: In the experiments, the performance gain compared to previous (truncated) transformer approaches are quite limited. I view the proposed method as a novel position encoding mechanism, expecting to see the comparison of it against vanilla/truncated Transformers with more advanced position encodings, such as Rotary embeddings, ALiBi and/or Transformer-XL.

**Questions:**

I feel more confused than amazed about the fact that while SSM alone cannot build a successful language model, using it (as a replacement of the position encoding) along with truncated (windowed/chunked) Transformer will simply result in an efficient long-term dependency capturing mechanism. I would appreciate it if the authors can conduct some ablation study to show that the inferior versions of SSM are, while still computationally efficient, not capable enough to support the dependency-capturing capabilities in language modeling, compared to the proposed SPADE.

---

> ### Author Response · Authors · 2023-11-23
>
> Weaknesses:
>
> * We conduct additional experiments to demonstrate the effectiveness of SPADE. Specifically, we use S4, S5 [2], and Hyena [1] as the SSM in SPADE (Figure 3). We remark that other SSMs, such as GSS [3], are also outperformed by state-of-the-art models: S5 and Hyena. We also compare SPADE with positional embedding methods such as relative positional embedding and rotary positional embedding. We conduct language modeling experiments, and we report the ppl on the test set (lower the better). All models in the experiments have approximately 120M parameters.
>
> * From the results, we can see that SPADE is easily extendable to accommodate various SSM. Moreover, for all configurations, we see that SPADE significantly outperforms the SSM-only model.
>
> Questions:
>
> We explained the reason SSMs cannot handle local dependency well in Section 3.1. In language modeling, local information is more important than global information (in contrast to the Long Range Arena benchmark, which is specifically designed for long-range dependence modeling). This is because in practice, we rarely see words (tokens) that are thousands of positions apart exhibit strong dependencies [4]. Therefore, the results in Figure 2 demonstrates that SSMs cannot handle local information as well as attention models.
>
>
> |  | ppl |
> |---|---|
> | Transformer | 22.3 |
> | Transformer-Relative | 20.9 |
> | Transformer-Rotary | 20.8 |
> | S4 | 22.7 |
> | SPADE-S4 | 20.2 |
> | S5 | 23.0 |
> | SPADE-S5 | 20.0 |
> | Hyena | 22.9 |
> | SPADE-Hyena | 20.3 |
>
> [1] https://arxiv.org/pdf/2302.10866.pdf \
> [2] https://arxiv.org/pdf/2208.04933.pdf \
> [3] https://openreview.net/forum?id=5MkYIYCbva \
> [4] https://arxiv.org/pdf/1905.07799.pdf

---

### Official Review · Reviewer_o9d9 · 2023-11-01

**Soundness:** 4 excellent
**Presentation:** 4 excellent
**Contribution:** 3 good
**Rating:** 8
**Confidence:** 4

**Summary:**

The paper introduces State sPace AugmenteD TransformEr (SPADE) which combines S4 with local attention to achieve and transformer model that avoids the quadratic sequence length scaling.

The authors note that local attention usually hurts the model's ability to attend to long-range dependencies and state-space models (SSM) usually do poorly in tasks where local information is important like language modeling. By having the bottom layer be a SSM and the rest of the layers perform local attention, they get the best of both worlds.

This is evidenced by performance in Long Range Arena (LRA), language modeling, and GLUE.

**Strengths:**

The empirical results are both thorough and impressive. The authors did many ablations and on many different tasks and the model performs well on all of them.

The paper is clear, and the idea is simple and intuitive. The paper is timed well to capture interest in large language models and context length.

Section 6.3 regrading the location and number of global layers anticipates many questions about the justification for the experiment setup.

**Weaknesses:**

Perhaps one ablation that wasn't done is length generalization. The authors claim that "our pre-trained model can extrapolate to any sequence length", and theoretical justification is sound, but it would be good to see empirical evidence.

Perhaps more configurations could be tried like different attention mechanisms for different heads or alternating layers.

**Questions:**

Does the method scale further? Is it easy to parallelize on multiple devices or even longer sequence lengths?

Were there any investigations into what type of global information is being propagated. Perhaps by looking at attention scores?

---

> ### Author Response · Authors · 2023-11-23
>
> Weaknesses:
>
> 1. One of the properties of SSMs is that they do not need to be trained to capture global information (although training usually brings more gain). In our pre-training experiments, we do not train the SSM in the bottom layer, and we only train the attention and FFN weights. Therefore, even though we set the pre-training length to 512, our pre-trained model can handle any sequence length during fine-tuning, e.g., the summarization tasks we considered.
>
> 2. We conduct ablation experiments on the location and number of SSMs in Section 6.3. We conduct additional experiments to demonstrate the effectiveness of SPADE. Specifically, we use S4, S5 [2], and Hyena [1] as the SSM in SPADE (Figure 3). We remark that other SSMs, such as GSS [3], are also outperformed by state-of-the-art models: S5 and Hyena. We also compare SPADE with positional embedding methods such as relative positional embedding and rotary positional embedding. We conduct language modeling experiments, and we report the ppl on the test set (lower the better). All models in the experiments have approximately 120M parameters. From the results, we can see that SPADE is easily extendable to accommodate various SSM. Moreover, for all configurations, we see that SPADE significantly outperforms the SSM-only model.
>
> Questions:
>
> 1. Our model is easier to scale compared with SSM-only models. In our experiments, we find that using a single SSM in the bottom layer suffices for the model to perform well. Therefore, the majority of SPADE are attention-based layers, which are easy to scale.
>
> 2. Thank you for the suggestion. We will further investigate how SSMs propagate global information. We would like to highlight that in SPADE, we use the SSM to model global information, and we use the local attention modules to capture more fine-grained local information,
>
> |  | ppl |
> |---|---|
> | Transformer | 22.3 |
> | Transformer-Relative | 20.9 |
> | Transformer-Rotary | 20.8 |
> | S4 | 22.7 |
> | SPADE-S4 | 20.2 |
> | S5 | 23.0 |
> | SPADE-S5 | 20.0 |
> | Hyena | 22.9 |
> | SPADE-Hyena | 20.3 |
>
>
> [1] https://arxiv.org/pdf/2302.10866.pdf \
> [2] https://arxiv.org/pdf/2208.04933.pdf \
> [3] https://openreview.net/forum?id=5MkYIYCbva

---

### Official Review · Reviewer_6SGN · 2023-11-02

**Soundness:** 3 good
**Presentation:** 4 excellent
**Contribution:** 2 fair
**Rating:** 5
**Confidence:** 4

**Summary:**

This paper presents a novel model architecture designed to handle the complexities of both long-sequence processing. This is achieved by integrating S4 global attention with local attention mechanisms to form a hierarchical structure. The S4 layer is utilized at the bottom to capture long dependencies, while the local attention layers above aim to simplify attention complexity and expedite computation. The model outperforms the S4 alone and traditional Transformers in specific tasks.

**Strengths:**

Strengths:
1. This paper describes a novel model that integrates S4 global attention and local (window-based or chunk-based) attention to address both long-range dependencies in language modeling. The hierarchical structure, with S4 at the bottom and local attention on top, aims to balance complexity and computation speed.
2. The model has shown improvements over S4 and traditional Transformers in long-range and text generation tasks.

**Weaknesses:**

Weaknesses:
1. S4 and Transformer are widely-used models, combining them together brings somehow incremental novelty contributions.
2. The comparisons with alternative methods for capturing global context, such as RNN-like mechanisms or other efficient attention mechanisms, are incomplete and lack essential details. Only S4 is compared in table 1.  Some recent long-sequence modeling studies:
  a. LongNet: Scaling Transformers to 1,000,000,000 Tokens
  b. Long Range Language Modeling via Gated State Spaces

**Questions:**

Questions:
1. T5-base is not implemented with the same setting with the proposed model. It would be more convincing to report apple-to-apple comparisons with the same setting, like the same pre-trained datasets.

---

> ### Author Response · Authors · 2023-11-23
>
> Weaknesses:
>
> 1. The major contribution of the SPADE is its scalability and extensibility. Existing works on SSMs need significant engineering efforts to scale up. However, by combining SSMs with Transformer layers, the proposed method is easy to scale and can be easily extended to accommodate advanced SSM structures.
>
> 2. We address the following about baselines:
>     * From the Hyena paper [1], RNN-like mechanisms such as RWKV have worse performance than SSM such as Hynea. We conduct additional experiments to demonstrate the effectiveness of SPADE. Specifically, we use S4, S5 [2], and Hyena [1] as the SSM in SPADE (Figure 3). We remark that other SSMs, such as GSS [3], are also outperformed by state-of-the-art models: S5 and Hyena. We also compare SPADE with positional embedding methods such as relative positional embedding and rotary positional embedding.We conduct language modeling experiments, and we report the ppl on the test set (lower the better). All models in the experiments have approximately 120M parameters.
>
>     * From the results, we can see that SPADE is easily extendable to accommodate various SSM. Moreover, for all configurations, we see that SPADE significantly outperforms the SSM-only model.
>
> Questions:
>
> 1. We pre-train a T5 model using the same setting as SPADE, and we include the fine-tuning results in Table 3. From the results, we see that the re-implemented T5 model outperforms the public T5 model. However, SPADE still significantly outperforms the re-implemented T5 model (86.8 vs 85.1 for average score on GLUE).
>
>
> |  | ppl |
> |---|---|
> | Transformer | 22.3 |
> | Transformer-Relative | 20.9 |
> | Transformer-Rotary | 20.8 |
> | S4 | 22.7 |
> | SPADE-S4 | 20.2 |
> | S5 | 23.0 |
> | SPADE-S5 | 20.0 |
> | Hyena | 22.9 |
> | SPADE-Hyena | 20.3 |
>
>
> [1] https://arxiv.org/pdf/2302.10866.pdf \
> [2] https://arxiv.org/pdf/2208.04933.pdf \
> [3] https://openreview.net/forum?id=5MkYIYCbva

---

### Official Review · Reviewer_QKEi · 2023-11-03

**Soundness:** 3 good
**Presentation:** 3 good
**Contribution:** 3 good
**Rating:** 6
**Confidence:** 2

**Summary:**

The paper proposes the SPADE (State sPace AugmenteD TransformEr) model, which augments a State Space Model (SSM) to a transformer model to effectively capture global information from long sequences. It also leverages local attention modules to capture local information. Both SSN and local attention can be computed efficiently compared with full attention mechanisms. The paper presents extensive experimental results and conducts ablation studies to show the effectiveness of the proposed approach.

**Strengths:**

- The paper is well written and easy to understand.
- The proposed architecture strikes a balance between simplicity and complexity, demonstrating strong performance on sequences of varying lengths while incurring lower computational expenses compared to full attention.
- Extensive experiment results across diverse datasets and tasks, along with ablation studies are provided.

**Weaknesses:**

- The paper primarily consolidates existing concepts, such as SSM and local attention, and offers limited novelty in terms of new methodologies.
- Aside from the experimental results, it falls short in providing a comprehensive understanding of why this architecture is effective, and more crucially, in identifying scenarios where this approach may not be as effective.

**Questions:**

- In Figure 3, you simply concatenate SSM and local attention output, and apply a weight $\bf W$, did you try any other method to fuse them?
- It would be beneficial to include a relatively rigorous complexity analysis of SPADE comparing with various other methods, potentially in the appendix.

---

> ### Author Response · Authors · 2023-11-23
>
> Weaknesses:
>
> * One of the main contributions of SPADE is its scalability and extensibility. Existing works on SSMs need significant engineering efforts to scale up. However, by combining SSMs with Transformer layers, the proposed method is easy to scale and can be easily extended to accommodate advanced SSM structures.
>
> * In Figure 6, we systematically study why the proposed architecture is effective. Specifically, through extensive experiments, we demonstrate that using a single global layer (i.e., a layer augmented with SSM) at the bottom of the model suffices for the model to perform well.
>
> * We would like to highlight that existing models with SSMs focus on long-sequence modeling. However, through our experiments in Table 3, we demonstrate that SPADE is effective for both long-sequence and short-sequence modeling. This compensates a drawback of existing methods.
>
> Questions:
>
> * We also investigated other options: in Figure 6 we investigate the location of the SSM, and we conclude that the best location is to put the SSM at the bottom of the model. We also investigated other options to combine attention with LLM, such as stacking the SSM with attention (e.g., first compute the SSM, and then the output is fed to the attention). In practice, we only find marginal differences between this approach and the proposed method.
>
> * For SSM such as S4 and Hyena, the complexity is $O(L \log (L))$, where $L$ is the sequence length. For a model with $N$ SSM layers, the total time complexity is $O(N L \log (L))$. In SPADE, we only use one SSM layer, and the other layers contain local attention modules with linear complexity. Therefore, the overall complexity of SPADE is $O(L \log (L) + (N-1) L)$, which is faster than models with only SSM layers. We demonstrate this in Figure 5.

---

### Meta-Review · Area_Chair_Dbmd · 2023-12-13

**Metareview:**

Reviewers find the proposal to combine SSMs with Local attention simple and effective. The main concerns are around limited comparisons to other SSMs, other Efficient Attention approaches and more diverse/long context tasks. Authors presented additional results during response showing this approach combines with other SSMs as well. However the other questions and comparisons still need to be addressed. Given the simplicity of the ideas, more comparisons and baselines will make the paper stronger and results more convincing. Hence I suggest reject for current submission but encourage to submit for future venues addressing reviewer concerns.

**Justification For Why Not Higher Score:**

Missing comparisons and baselines

**Justification For Why Not Lower Score:**

N/A

---

### Decision · Program_Chairs · 2024-01-16

Reject